# Direct High-Power Microwave Interaction with a Zinc Wire: A Novel Route to Crystalline ZnO Nanopowders Synthesis

**DOI:** 10.3390/ijms26188981

**Published:** 2025-09-15

**Authors:** George Mogildea, Marian Mogildea, Sorin I. Zgura, Natalia Mihailescu, Doina Craciun, Valentin Craciun, Oana Brincoveanu, Alexandra Mocanu, Vasilica Tucureanu, Cosmin Romanitan, Alexandru Paraschiv, Bogdan S. Vasile, Catalin-Daniel Constantinescu

**Affiliations:** 1Institute of Space Science—INFLPR, 077125 Magurele, Romania; george_mogildea@spacescience.ro (G.M.); szgura@spacescience.ro (S.I.Z.); 2National Institute for Laser, Plasma and Radiation Physics, 077125 Magurele, Romania; natalia.serban@inflpr.ro (N.M.); doina.craciun@inflpr.ro (D.C.); 3Extreme Light Infrastructure for Nuclear Physics, 077125 Magurele, Romania; 4National Institute for Research and Development in Microtechnologies, 077190 Voluntari, Romania; oana.brincoveanu@imt.ro (O.B.); alexandra.mocanu@imt.ro (A.M.); vasilica.tucureanu@imt.ro (V.T.); cosmin.romanitan@imt.ro (C.R.); 5Faculty of Chemical Engineering and Biotechnologies, National University of Science and Technology Politehnica Bucharest, 011061 Bucharest, Romania; 6Romanian Research and Development Institute for Gas Turbines, 061126 Bucharest, Romania; alexandru.paraschiv@comoti.ro; 7National Research Center for Micro and Nanomaterials, Bucharest National Polytechnic University of Science and Technology, 060042 Bucharest, Romania; vasile_bogdan_stefan@yahoo.com; 8Research Center for Advanced Materials, Products and Processes, Bucharest National Polytechnic University of Science and Technology, 060042 Bucharest, Romania; 9Aix-Marseille Université, CNRS, LP3 UMR 7341, 13009 Marseille, France; catalin.constantinescu@cnrs.fr

**Keywords:** microwaves, plasma, nanoparticles, ZnO

## Abstract

We present a novel approach for the synthesis of crystalline zinc oxide (ZnO) nanopowders based on the direct interaction of high-power microwave radiation with a zinc wire in atmospheric air. The process utilizes a localized microwave-induced plasma to rapidly vaporize the metal, followed by oxidation and condensation, resulting in the deposition of ZnO nanostructures on glass substrates. Plasma diagnostics confirmed the generation of a plasma in local thermodynamic equilibrium (LTE), characterized by high electron temperatures. Optical emission spectroscopy highlighted atomic species such as ZnI, ZnII, OI, OII, and NI, as well as molecular species including OH, N_2_ and O_2_. The spectral fingerprint of N_2_ molecules reveals the presence of high energy electrons, while the persistent occurrence of OI and OII emission lines throughout the plasma spectrum reveals that ZnO formation is mainly driven by the continuous dissociation of molecular oxygen. High crystallinity and chemical purity of the synthesized ZnO nanoparticles were confirmed through SEM, TEM, XRD, FTIR, and EDX characterization. The resulting nanorods exhibit a rod-like morphology, with diameters ranging from 12 nm to 63 nm and lengths between 58 nm and 354 nm. This low-cost, high-yield method offers a scalable and efficient route for metal oxide nanomaterial fabrication via direct metal–microwave coupling, providing a promising alternative to conventional physical and chemical synthesis techniques.

## 1. Introduction

Nanomaterials are widely used across diverse industrial sectors due to their enhanced chemical reactivity, which arises from their high surface-to-volume ratio [1]. Among them, zinc oxide (ZnO) has attracted significant attention for applications in agriculture, food packaging [2], biomedicine, and renewable energy, including solar cells, gas sensors, and photocatalytic hydrogen production [3,4,5]. ZnO offers several key advantages: it is low-cost, given zinc’s natural abundance [6], non-toxic, and exhibits strong photocatalytic and antibacterial properties [7]. To meet the increasing demand for ZnO nanomaterials, various synthesis techniques have been explored, such as wet chemical synthesis methods, biological methods, and plasma-assisted methods [8].

When considering chemical and biological synthesis methods for nanoparticle production, it becomes evident that both approaches come with specific limitations.

Chemical techniques, including sol–gel processing and hydrothermal or solvothermal synthesis, are known to yield high-quality nanoparticles. However, they often involve the use of toxic reagents or produce hazardous byproducts, posing risks to both human health and the environment. Moreover, these methods typically require extensive washing or purification steps, which can generate contaminated wastewater and present challenges for scaling up production efficiently. On the other hand, biological techniques such as virus-engineered nanoparticle synthesis or enzyme-mediated approaches are considered environmentally friendly, but they tend to lack precise control over particle size and morphology, which restricts their applicability in precision demanding fields [9]. To overcome the limitations associated with chemical and biological synthesis methods, researchers have increasingly turned to plasma-assisted techniques for nanoparticle production. These approaches aim to generate a wide variety of nanoparticles without releasing hazardous byproducts into the environment. As a result, several advanced methods and technologies have been developed, offering clean and efficient synthesis pathways. These include: Arc Plasma Synthesis (DC/RF), Inductively Coupled Plasma Synthesis, Spark Discharge Nanoparticle Generation, Laser Ablation Synthesis in Solution or Liquid, Plasma-Enhanced Chemical Vapor Deposition, Microwave Plasma Synthesis, and Microwave Plasma in Liquid Process. While these plasma-assisted methods are generally considered more sustainable compared to conventional chemical routes, they are not without their own drawbacks. Notable challenges include high energy requirements, increased operational costs, and bulky equipment configurations. Moreover, many techniques demand specific precursor materials and often face scalability issues for industrial level production. Additionally, certain plasma methods are affected by problems such as electrode degradation, limited control over particle morphology, or contamination of nanoparticles due to electrode erosion during synthesis [10,11,12,13,14,15,16,17,18,19]. These constraints highlight a clear need to develop alternative synthesis strategies that are scalable, energy efficient, low-cost, and non-toxic to the environment. Therefore, this paper introduces a novel, low-cost, high-yield microwave-assisted synthesis method for producing ZnO nanoparticles. The method operates under ambient air conditions eliminating the need for chemical precursors, solvents and process gas and offers a non-contact, plasma-based route for nanoparticle generation.

The fundamental distinction between conventional microwave-assisted synthesis and the method described in this work lies in the mechanism by which plasma is generated. In traditional systems, microwaves interact with a process gas to generate plasma, followed by the introduction of a solid-phase precursor into the plasma zone. By contrast, the new approach employs the direct interaction between microwaves and a metal wire, allowing for the in situ synthesis of metal oxides directly from elemental metals. This approach has enabled the successful synthesis of high-purity metal oxide nanoparticles, such as In_2_O_3_, WO_3_, and TiO_2_, directly from elemental metal wires [20,21,22].

Although interest in microwave-assisted synthesis has steadily increased in recent years, the direct exposure of elemental metals to microwave radiation for the production of metal oxide nanoparticles remains largely unexplored. This gap in research is primarily rooted in the long-standing assumption that dielectric materials readily absorb microwave energy [23], whereas metals tend to reflect it [24]. However, recent experimental evidence has challenged this notion, demonstrating that under specific conditions, metallic powders and wires can strongly interact with microwave fields. These interactions enable energy absorption through mechanisms such as Eddy current induction or field emission, ultimately resulting in metal powders heating or plasma formation [25,26,27,28,29,30,31,32]. Therefore, this method relies on microwave-induced high electrical field and thermionic emission in a zinc wire to generate a localized plasma in atmospheric air, enabling zinc vaporization followed by in situ oxidation and deposition of ZnO nanoparticles on glass surfaces. Plasma characterization using optical emission spectroscopy technique revealed the presence of ZnI, ZnII, OI, OII and NI atomic species and OH, N_2_ and O_2_ molecular species, with electron temperatures reaching up to 11,400 K for ZnI and 6690 K for ZnII under local thermodynamic equilibrium (LTE) conditions.

Structural analyses using Scanning Electron Microscopy (SEM), Transmission Electron Microscopy (TEM), X-ray Diffraction (XRD), Fast Fourier-Transformed Infrared Spectroscopy (FTIR) analysis and Energy-Dispersive X-ray Spectroscopy (EDX) confirmed the formation of ZnO nanorods with high crystallinity and purity, averaging 24 nm in diameter and 155 nm in length.

This study introduces a fundamentally new concept for microwave-metal interaction as a synthesis route for oxide nanomaterials and opens a scalable pathway for environmentally friendly nanoparticle production using simple, widely available and inexpensive equipment.

## 2. Results

### 2.1. Optical Diagnostics of the Microwave Plasma

To characterize the plasma generated during the microwave-zinc wire interaction, optical emission spectroscopy (OES) was performed using an Ocean Optics USB 2000 spectrometer (Ocean Optics, Dunedin, FL, USA) (spectral resolution ~0.1 nm FWHM [33]). Due to the high plasma temperature, the lens of the optical fiber (connected to the spectrometer) was positioned 70 mm away from the plasma to prevent damage Spectral data were acquired with an integration time of 100 ms. Prior to data acquisition, the spectrometer was calibrated using a DH-mini UV–Vis-NIR Deuterium-Halogen Light Source from Ocean Optics, Inc. (Dunedin, FL, USA), ensuring precise wavelength accuracy and signal fidelity. Emission spectra were recorded with a 100 ms integration time and analyzed using Span V1.7 software [34]. The acquired emission spectrum, displayed in Figure 1, revealed distinct atomic and ionic lines spanning the UV–VIS-NIR range. Comparison with reference data from the NIST Atomic Spectra Database [35] and related literature [36] confirmed the presence of ZnI, ZnII, OI, OII and NI atomic species and OH, N_2_ and O_2_ molecular species.

### 2.2. Plasma Electron Temperature

The electron temperature (T_e_) was determined via the Boltzmann plot method [37], using selected emission lines of ZnI and ZnII (Table 1) present in the optical emission spectrum of the plasma.

To determine the electron temperature (T_e_) for ZnI and ZnII, the following equation was used:(1)$$ln\left(\frac{I\lambda }{gA}\right)=\;-\;\frac{E_{k}}{K_{B}T_{e}}\;-ln\left(\frac{4\pi Z}{hcN_{0}}\right)$$

To calculate the intensity gradient of each spectral line listed in Table 1, Equation (2) was used:(2)$$I=\frac{h\upsilon AN}{4\pi }=\frac{hcN_{0}gA}{4\pi Z}e^{-E_{k/K_{T}}}$$
where I—intensity of the transition, h—Planck’s constant, υ—frequency, g—degeneracy, A—transition probability, N—absolute number or number density, c—speed of light, N_o_—total species population, λ—wavelength of spectral lines emitted by microwave induced plasma, Z—partition function, E_k_—energy of the upper state of emission, K_B_—Boltzmann constant, T_e_—electron temperature and R^2^—statistical coefficient.

The Boltzmann plot could be fit with a linear Equation (3):(3)y=mx+b
where m is the slope. Equation (2) also takes a linear form, where y=ln(IgA), $x=E_{k}$, b=intercept and m=1KBT. The resulting linear fits (Figure 2a,b) yielded T_e_ values of approximately 11,400 K for ZnI and 6690 K for ZnII, respectively. The excellent linearity of the Boltzmann plots confirms that the plasma satisfies local thermodynamic equilibrium (LTE) conditions.

The clustering of data points at both ends of the Boltzmann plot for ZnII reflects the variable accuracy in measuring strong versus weak emission lines, primarily influenced by differences in the signal-to-noise ratio [38]. The surrounding nanoparticle cloud can attenuate the plasma emission through scattering and absorption [39], thereby reducing line intensity; nevertheless, the high electron temperature and the strong correlation coefficients (R^2^ > 0.9 for ZnI and ZnII) confirm the reliability of the results.

### 2.3. Plasma Electron Density

To determine the electron density (*n_e_*) in the microwave induced plasma the Saha-Boltzmann Equation (4) [40] was used.(4)ne=2(πm3kT)32h3·ImnI·Ai·giIIIijI·Amn·gmIexp−Eion+EjII−EmIkT
where
*m_e_*—mass of an electron, k—Boltzmann constant, h—Planck’s constant,T—plasma temperature, I_mn_—intensity of the line transition from m-level to *n*-level,A_mn_—Einstein coefficient of the transition probability for spontaneous transition,$g_{m}^{I}$—degeneracy of the upper level, $E_{m}^{I}$—energy of the upper energy level of an atomic line, E_ion_—1st ionization energy, I_ij_, A_ij_, $g_{i}^{II}$ and $E_{i}^{II}$ are the corresponding parameters of an ionic line.

After applying Equation (4) to the emission parameters of the microwave-induced plasma, the value of n_e_ was estimated at around 5.9 × 10^16^ cm^−3^.

### 2.4. Optical Characterization of the ZnO Nanoparticles

To evaluate the optical bandgap of ZnO nanoparticles produced by the direct interaction of microwaves with a Zn wire in atmospheric air, the nanoparticles were deposited onto a quartz substrate and their optical transmittance spectra (250 nm–850 nm) were recorded with a fiber-optic spectrometer (Ocean Optics USB2000) from Ocean Optics, Inc. (Dunedin, FL, USA) coupled to a DH-mini UV–Vis-NIR Deuterium-Halogen Light Source, then the bandgap was then determined by Tauc analysis [41]. In Figure 3 is displayed the optical transmittance spectrum of the ZnO and in Figure 4 is displayed the Tauc plot for ZnO (direct-allowed transition).

Analyzing Figure 3, it is observed that transmittance of the ZnO powder increases toward longer wavelengths; this trend is typical of strongly scattering media. The small fluctuations in the transmittance (Figure 3 ZnO raw) are consistent with a porous [42] and polydisperse powder layer.

The Tauc plot (direct-allowed) indicates an Eg ~3.1 eV band gap for the porous ZnO layer, which is lower than pure ZnO (Eg = 3.2 eV–3.4 eV). This lower value caused by sub-gap absorption from defects (oxygen vacancies, Zn interstitials) and strong scattering in the porous nanopowder, which broaden the absorption edge and bias the Tauc fit downward [43].

### 2.5. Structural Analysis of ZnO Nanoparticles

Structural characterization of the ZnO powders was carried out using a Nova NanoSEM 630 scanning electron microscope (FEI Company, Hillsboro, OR, USA) operated at an accelerating voltage of 10 kV. SEM analysis confirmed the nanoscale dimensions of the particles, revealing the presence of both nanoparticles (NPs) and nanorod-like domains (NRDs).

Elemental composition was evaluated using energy-dispersive X-ray spectroscopy (EDX, Figure 5) mapping, which identified zinc (Zn) and oxygen (O) as the primary constituents being uniformly distributed on the glass substrate. Quantitative analysis of particle size distribution was conducted by evaluating approximately 200 individual NPs and NRDs from SEM images using ImageJ software version 1.54 (NIH). The resulting size histograms (Figure 3) were best fitted with a Gaussian function. As shown in Table 2, the ZnO NPs ranged from 12 to 63 nm, with a full width at half maximum (FWHM) of 15–32 nm and a mean diameter of 24 nm±9 nm. In contrast, the NRDs exhibited a broader size range of 58 to 354 nm, a FWHM of 96 nm–218 nm, and a mean size of 155 nm±57 nm, highlighting the morphological heterogeneity within the ZnO sample.

The XRD analysis, performed with an Empyrean instrument operating at a power of 45 kV and 40 mA with a Cu anode showed that during microwaves discharge crystalline ZnO nanoparticles were formed. In Figure 6 the XRD pattern is displayed acquired from ZnO powders.

The XRD patterns present a couple of intense and narrow diffraction peaks at 2θ = 31.63°, 34.31°, 36.23°, 47.47°, 56.62°, 62.46°, 66.28°, 67.82°, and 69.0°. These were unambiguously attributed to (100), (002), (101), (102), (110), (110), (103), (200), (112) and (201) reflections of wurtzite ZnO with a = b = 0.32 nm, c = 0.52 nm. The crystal quality was evaluated considering the size of the crystalline domains (mean crystallite size), τ by the well-known Scherrer Equation (5), that gives the relation between the crystalline domain size and peak broadening:(5)$$\beta :\tau =\;\frac{k\lambda }{\beta cos\theta }\;,$$
where k is a dimensionless factor that represents the shape factor of the crystallites and it varies between 0.62 and 2.08 depending on the crystallite geometry and distribution [44] and θ is the angular position of the diffraction peak. Based on the analysis of the (101) XRD diffraction peak and the application of the Scherrer equation (Equation (4)), in the assumption of rod-like shape for crystallite shape [45] the average crystallite size of ZnO was determined to be approximately 25.6 nm, which is very similar to the diameter of the crystals estimated from SEM images. TEM analysis (Figure 7a–c) completes structural analyses of the ZnO nanoparticles. Bright-Field Transmission Electron Microscopy (BFTEM), High-Resolution TEM (HRTEM), and Selected Area Electron Diffraction (SAED) images were obtained using a Titan Themis transmission electron microscope (Thermo Fisher Scientific, Waltham, MA, USA), equipped with aberration correction and operated at an accelerating voltage of 200 kV. For sample preparation, a small amount of nanopowder, due to low mechanical adhesion, was easily dislodged from the substrate following deposition and dispersed in deionized water and sonicated for 5 min.

Then, 10 µL of the resulting suspension was dropped onto a 400-mesh lacey carbon-coated copper grid and allowed to dry prior to imaging.

To validate the structural and morphological findings obtained from XRD and TEM analyses, and to identify potential surface-bound species or secondary phases that could indicate the presence of impurities FTIR spectroscopy was employed. Using the Vertex 80 V spectrometer (Bruker Optics, Billerica, MA, USA), equipped with a grazing angle specular reflectance accessory, analyses of the oxide film were performed, at an incident angle of 45°.

Vibrational spectrometry of the ZnO sample deposited on a glass substrate, revealed a spectrum characterized by high-intensity absorption bands associated with the substrate, and low-intensity bands that confirm the existence of ZnO on the glass.

Figure 6, recorded in the 4000 cm^−1^–380 cm^−1^ range provides the complete FTIR spectrum of ZnO deposited on the glass substrate, while Figure 8b presents an enlarged view of the 500 cm^−1^–380 cm^−1^ region, highlighting the characteristic Zn-O peaks.

The characteristic bands of the ZnO crystal structure (Figure 8b) can be observed at 410 cm^–1^, 391 cm^−1^ and 388 cm^−1^. The presence of more absorption bands assigned to Zn-O bonds is due to multiple size or the coexistence of different types of morphologies.

The other bands observed in the spectrum (Figure 8a) are attributed to the vibration mode of the O-H (3675 cm^−1^), B-O (1378 cm^−1^), P-O (541 cm^−1^, 4212 cm^−1^, 400 cm^−1^) and Si-O (1300 cm^−1^–800 cm^−1^) bonds present in the glass. These features were assigned to the substrate by comparison with the corresponding reference FTIR spectrum.

The FTIR spectrum of the ZnO sample deposited on glass exhibits absorption bands that correspond well with the characteristic group frequencies described in Infrared and Raman Characteristic Group Frequencies by G. Socrates [46], confirming the reliability of the spectral assignments.

No organic contamination bands or absorption bands corresponding to C-O stretching vibrations were detected in the FTIR spectrum. This absence is expected, since the synthesis involved only the direct oxidation of metallic zinc wire under microwave irradiation in ambient air, without any carbon-containing precursors. The result confirms that the obtained ZnO nanostructures are free of carbon-oxygen impurities.

## 3. Discussions

Nowadays, microwave technology plays a crucial role in generating plasma [47], which is essential for an increasing range of applications.

In recent years, advanced systems have emerged for nanomaterial synthesis, enabling the production of nanoparticles from chemical precursors or metallic powders through the generation of hot plasma (T_e_ ~6500 K and n_e_ = 10^13^ cm^−3^) in a process gas [48,49]. In these setups, the raw material is injected into the plasma, vaporized, and then condensed into nanoparticles. In contrast, the microwave synthesis method presented here distinguishes itself through a remarkably simple experimental setup, excellent energy efficiency, and a significantly shorter reaction time. It operates without the need for controlled environments or carrier gases. Materials and methods shows a cloud of ZnO nanoparticles generated after just 10 s of microwave exposure at 650 W, during which approximately 42 mg of ZnO was generated. The structural characteristics of the ZnO nanoparticles, confirmed by TEM, SEM, and XRD analyses, indicate that their morphology is comparable to those produced by conventional microwave-based methods. However, the key advantage of the present technique lies in its significantly higher production yield, approximately 15 g/hour, compared to the 4 g/h typically achieved using conventional systems [50]. The significantly higher yield is primarily due to the elevated plasma temperature [51] achieved by this technique, reaching approximately 11,600 K and n_e_ = 5.9 × 10^16^ cm^−3^. This high temperature arises from the system’s unique operating principle, instead of relying on the interaction between microwaves and a process gas, as in conventional methods, the microwaves in this setup interact directly with a zinc wire in ambient air. Without the presence of the metal wire, plasma ignition does not occur under atmospheric conditions. This direct interaction results in the formation of a highly energetic plasma, which facilitates rapid and efficient nanoparticle synthesis. Moreover, unlike traditional systems where the presence of atmospheric air can lead to contamination of the nanoparticles [52], the novel configuration employed here leverages ambient air [53] to enhance plasma temperature without compromising the purity of the final product.

Therefore, the formation of ZnO under these conditions follows a complex multi-step mechanism. This involves plasma generation, dissociation of O_2_ molecules, vaporization of zinc, and its subsequent oxidation.

This complexity arises because the energy of microwave photons (~10^−5^ eV) is insufficient to directly initiate molecular dissociation or produce reactive species. As a result, energetic electrons required for these transformations are generated through a combination of field emission and thermionic emission. Initially, high-intensity microwave irradiation induces field emission of low-energy electrons (0.3 eV–1 eV) [54] from the Zn surface. These electrons are then accelerated within the microwave field, colliding with gas-phase molecules and the Zn wire itself, resulting in localized heating. This thermal effect promotes thermionic emission, and together, the two mechanisms create a hybrid thermo-field emission regime [55] that generates energetic electrons. These electrons, through sustained acceleration, become capable of dissociating O_2_ and N_2_ molecules and ionizing Zn vapors. These crucial plasma-phase reactions are evidenced by the atomic species ZnI, ZnII, OI, OII, and NI, as well as molecular species like OH, O_2_, and N_2_, ref. [56] observed in the optical emission spectrum (Figure 1).

The presences of OH molecular bands indicate residual atmospheric humidity during synthesis. Spectroscopic analysis reveals that the plasma contains electrons with energies up to 15 eV. Two complementary emission fingerprints support this: the presence of N_2_ Second Positive System bands $\left(C^{3}\Pi_{u}\to B^{3}\Pi_{g}\right)$, particularly the (0-0) and (0-2) vibrational transitions at 337.1 nm and 357.7 nm, and the widespread appearance of atomic and ionic oxygen lines (OI and OII) across the emission spectrum [57]. Atomic oxygen is formed through two primary mechanisms: direct electron impact dissociation of O_2_, requiring about 5.1 eV to break the O = O bond [58], and energy transfer from microwave-excited N_2_ molecules (e.g., $C^{3}\Pi_{u}$ state, ~11 eV) [59], which dissociate O_2_ via collisional quenching or resonant energy transfer. This second pathway accounts for up to 30% of the total discharge energy being converted into fast gas heating in air [60]. In contrast, when pure O_2_ is used, the absence of these mechanisms results in lower heating efficiency [61]. Although ZnO formation may also involve reactions with excited molecular oxygen (O_2_*), the dominant pathway is the direct recombination of Zn vapors with atomic oxygen, as atomic oxygen is more reactive and abundant in the plasma [62]. This mechanism contributes to the high purity and crystallinity of the ZnO powder, as shown by the Selected Area Electron Diffraction (SAED) pattern (Figure 7c), which confirms the presence of a single ZnO phase with a hexagonal wurtzite crystal structure, consistent with the XRD pattern (Figure 6). The SAED pattern displays concentric diffraction rings, each corresponding to specific crystallographic planes, identified by their respective Miller indices. The diffuse and broad nature of the rings particularly near the center suggests the presence of small crystallites, which is typical for nanoscale materials. Further insight is provided by the Fast Fourier Transformed (FFT) analysis of an elongated nanoparticle. The FFT pattern reveals the presence of nearly all major Miller indices characteristic of wurtzite ZnO, even though such elongated crystals are typically expected to exhibit a dominant orientation. This observation is corroborated by the XRD data, which shows no preferential orientation among the main diffraction peaks, supporting the conclusion that the ZnO crystallites are randomly oriented, as also evidenced by the TEM image in Figure 7b. The high crystallinity of the nanoparticle is evident, with no detectable amorphous phase at the edges. The inset highlights well-resolved atomic columns aligned along the (103) and (110) crystallographic directions, corresponding to interplanar spacings of 1.03 Å and 1.62 Å, respectively.

TEM analysis of the samples shown in Figure 7b,c reveals that the microwave induced plasma synthesis method produces ZnO nanoparticles with a variety of morphologies. Statistical analysis of approximately 200 features from SEM images revealed that the ZnO sample consisted predominantly of nanorod-like (NRD) domains. The morphological yield of NRDs was ~87% (155 ± 57 out of 179 features), while nanoparticles accounted for only ~13% (24 ± 9 out of 179). These results demonstrate that the synthesis method strongly promotes anisotropic growth along the c-axis, favoring the preferential formation of nanorods.

Although this synthesis method can introduce native point defects in ZnO (oxygen vacancies and zinc interstitials), it remains attractive because it is simple, low-cost, and high yield. For semiconductor or photocatalytic applications, a subsequent thermal treatment (annealing/calcination) can reduce the defect density and increase the band gap Eg toward the bulk value [63].

To assess reproducibility, the synthesis was repeated in more than 40 independent deposition cycles carried out for different experimental purposes. In every case the process produced ZnO nanorods with consistent morphology, size distribution, and crystallinity. For this study, two representative samples were selected for detailed characterization. The uniform behavior of the synthesized ZnO across all cycles demonstrates that the method is highly reliable and readily reproducible.

Regarding the Zn wire vaporization yield: the Zn wire from waveguide focal point (~1 cm) were vaporized cleanly with no residue. Once that segment had been consumed and the vaporization front moved away from the focal spot, localized melting occurred, and small molten beads formed along the wire and at the edge of the ceramic support.

Therefore, this nanoparticle synthesis method is limited by the absence of a mechanical system to feed the metal wire from a spool into the waveguide’s focal region and maintain the plasma discharge. Implementing such a system would allow for a continuous nanoparticle production.

## 4. Materials and Methods

To develop an accessible and scalable method for ZnO nanomaterial synthesis, a custom, microwave plasma reactor based on components employed in commercial microwave ovens was constructed. The system utilizes a magnetron (frequency = 2.45 GHz, power of the microwaves = 800 W) coupled to a TM_011_ cylindrical copper waveguide (10.5 cm diameter × 11 cm length), optimized to generate a high-intensity microwave field (~8 MW/cm^2^ at its first resonant node, located on the cavity’s cylindrical axe at 6.5 cm away from the antenna. Figure 9 shows the schematic of the setup.

A high-purity (99.9%) zinc wire (5 cm long, 0.5 mm diameter) was inserted with one end positioned precisely in this focal zone. Figure 10 presents the EDX spectrum of the raw Zn wire prior to vaporization, confirming its high purity.

Microwave plasma ignition was initiated at a microwave power of 800 W, followed by a power decrease to 650 W using a custom electronic control module that controlled the vaporization rate. Plasma ignition was not sustained below 500 W. Figure 11 displays the metallic nanoparticles generation during the discharge process.

Upon microwave irradiation, the Zn wire was rapidly vaporized as a result of localized plasma generation, producing Zn vapor that subsequently oxidized in ambient air and condensed both on the inner walls of the waveguide and on a vertically mounted glass substrate positioned directly above the plasma region. An exposure time of 10 s at 650 W was sufficient to consume ~2.7 cm of zinc wire and generate a visible cloud of ZnO nanoparticles (Figure 12). This process yielded a uniform nanoparticle layer with an estimated average thickness of 3 μm to 5 μm on the glass substrate.

The microwave plasma generator operates within a sealed pressure chamber, where the zinc wire is vaporized and subsequently oxidized to form ZnO nanoparticles. Only after nanoparticle condensation is complete is the chamber opened to collect the deposited substrate, ensuring safe operation without exposure to zinc vapor.

## 5. Conclusions

This study demonstrates a novel and cost-effective route for synthesizing high-purity, crystalline ZnO nanoparticles by directly exposing a zinc wire to high-power microwave irradiation under ambient air conditions. Structural and morphological characterizations (SEM, TEM, XRD, FTIR, and EDX) confirmed the formation of ZnO nanorods with diameters ranging from 12 nm to 63 nm and lengths between 58 nm and 354 nm. Plasma diagnostics revealed that the microwave discharge generates both atomic and molecular species under LTE conditions, with spectroscopic analysis indicating electrons energies up to 15 eV. Excited N_2_ molecules were found to play a key role in dissociation of O_2_, while the dominant formation pathway involves the direct recombination of Zn vapors with atomic oxygen, owing to the higher reactivity and abundance of the latter. This mechanism accounts for the high purity and crystallinity of the resulting ZnO powder, which remain unaffected even in the presence of water vapor in atmospheric air. Furthermore, the porous ZnO layer exhibited a slightly reduced band gap (~3.1 eV) compared to bulk ZnO (~3.3 eV), a difference attributed to native defects and scattering effects. Nevertheless, the synthesis method remains highly attractive due to its simplicity, low cost, and high yield, while post-annealing offers a means of reducing defect density and restoring the band gap closer to the bulk value.

## Figures and Tables

**Figure 1 ijms-26-08981-f001:**
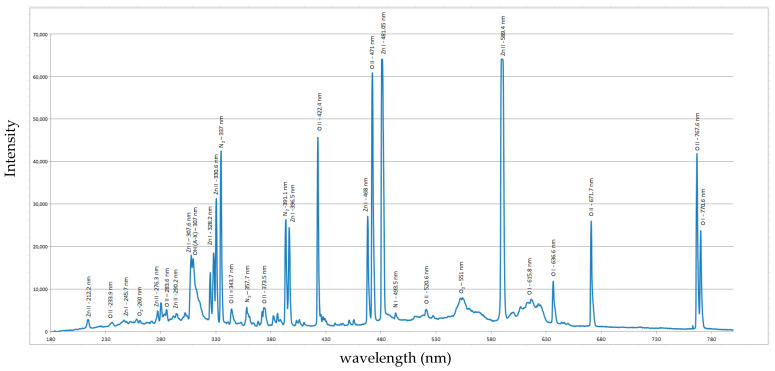
Optical emission spectrum of the plasma created at 650 W microwave power in atmospheric air.

**Figure 2 ijms-26-08981-f002:**
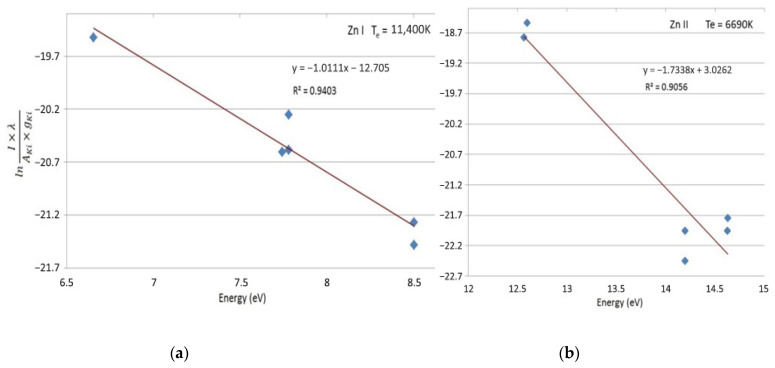
(**a**) Boltzmann Plot for ZnI. (**b**) Boltzmann Plot for ZnII. T_e_—electron temperature.

**Figure 3 ijms-26-08981-f003:**
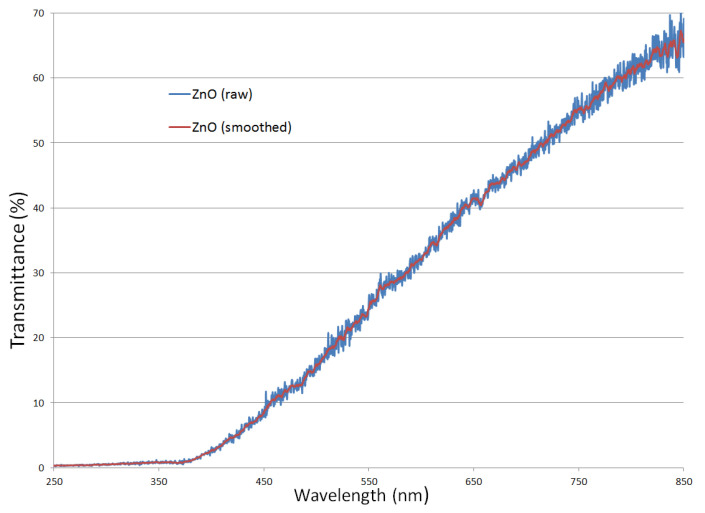
The optical transmittance spectrum of the ZnO.

**Figure 4 ijms-26-08981-f004:**
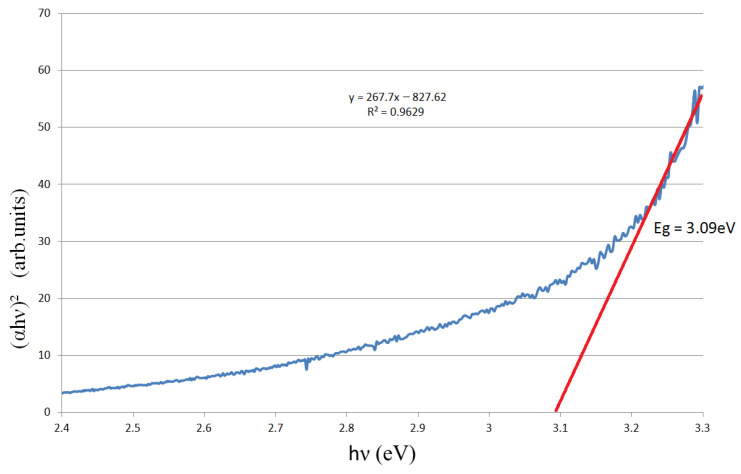
Tauc plot for ZnO (direct-allowed transition). Eg—optical band gap energy. The blue curve shows the experimental data, the red line is the linear fit used to extract Eg.

**Figure 5 ijms-26-08981-f005:**
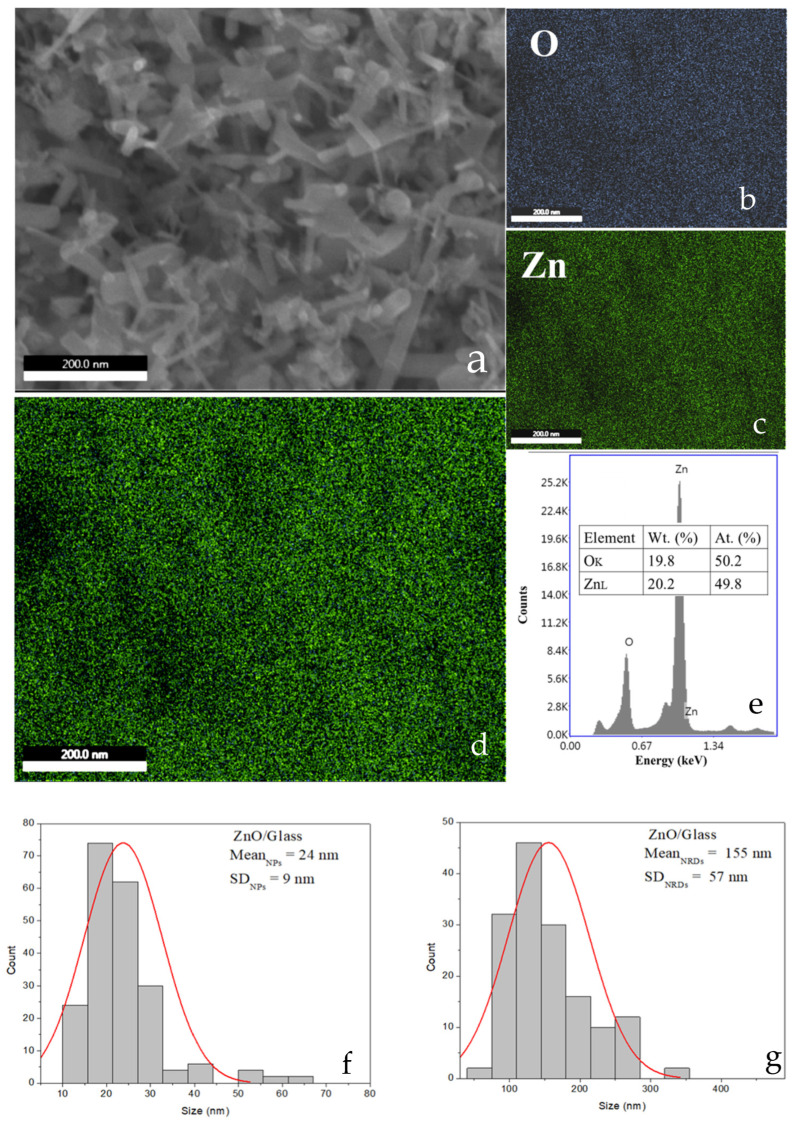
SEM image and EDX elemental analysis of the ZnO layer deposited on glass. (**a**) SEM micrograph of the ZnO layer; (**b**,**c**) EDX elemental maps of O and Zn, respectively; (**d**) superimposed elemental maps confirming the homogeneous presence of ZnO; (**e**) EDX analysis reveals a Zn:O ratio close to 1:1, consistent with ZnO stoichiometry, (**f**) Size distribution of ZnO nanoparticles; (**g**) Size distribution of ZnO nanorods. The histograms in (**f**,**g**) were fitted with Gaussian functions (red lines), confirming that the size distributions are approximately Gaussian. At.% denotes the atomic percent, Wt.% the weight percent, Mean_NPs_ the average nanoparticle size, Mean_NRDs_ the average nanorod size, and SD (standard deviation) indicates the spread of the size distribution.

**Figure 6 ijms-26-08981-f006:**
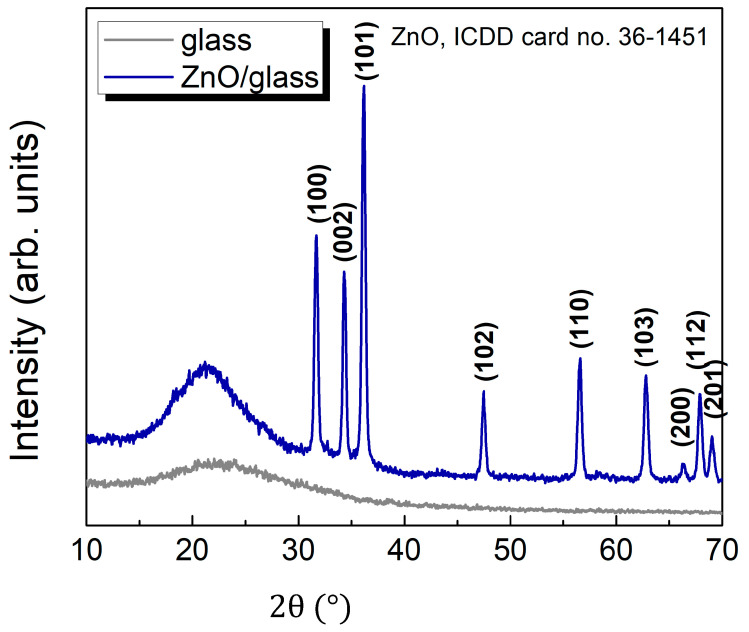
XRD patterns of the glass substrate (gray) and the ZnO layer deposited on glass (blue) plotted as a function of 2θ (°). The glass substrate shows a broad amorphous halo, while the ZnO/glass sample exhibits distinct diffraction peaks indexed to the hexagonal wurtzite ZnO phase (ICDD card no. 36-1451). ICDD refers to the International Centre for Diffraction Data.

**Figure 7 ijms-26-08981-f007:**
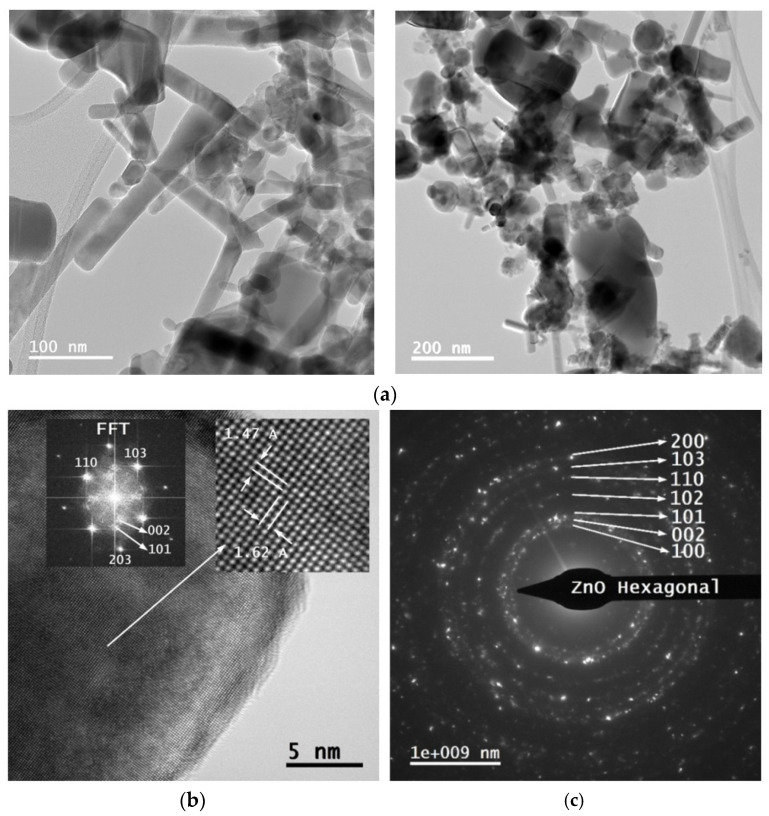
(**a**) BFTEM Images of the ZnO nanopowder at two different magnifications, (scale bar: 100 nm), clearly showing the nanorod-like domains. (**b**) HRTEM image of a ZnO nanorod. FFT—Fast Fourier Transform, Å (Angström), (**c**) SAED image of ZnO nanopowder.

**Figure 8 ijms-26-08981-f008:**
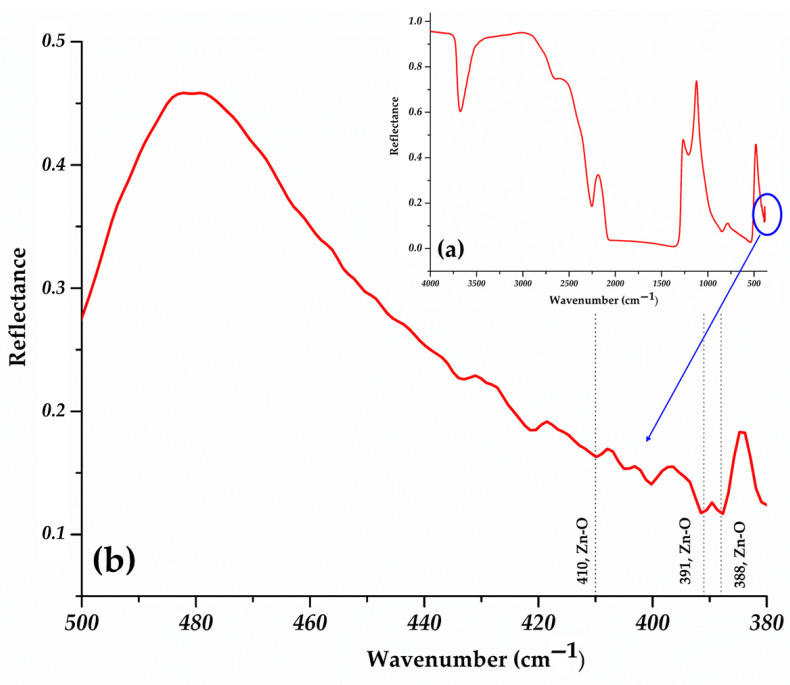
(**a**) FTIR spectrum of the ZnO nanoparticles deposited on the glass substrate. (**b**) FTIR spectrum of the ZnO nanoparticles. The blue circle and arrow indicate the region magnified in panel (**b**).

**Figure 9 ijms-26-08981-f009:**
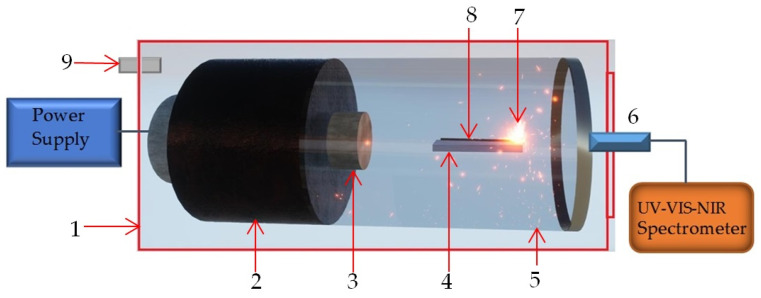
Design of the microwave device: 1—pressure chamber, 2—magnetron, 3—magnetron antenna, 4—ceramic support, 5—waveguide, 6—lens of the optical fiber, 7—plasma, 8—metallic wire, 9—gas connector.

**Figure 10 ijms-26-08981-f010:**
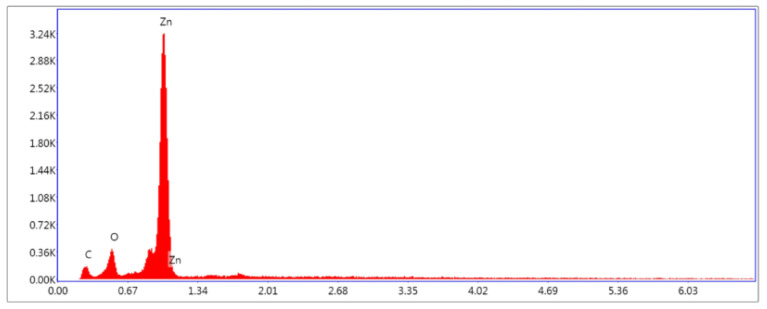
EDX analysis of the zinc wire.

**Figure 11 ijms-26-08981-f011:**
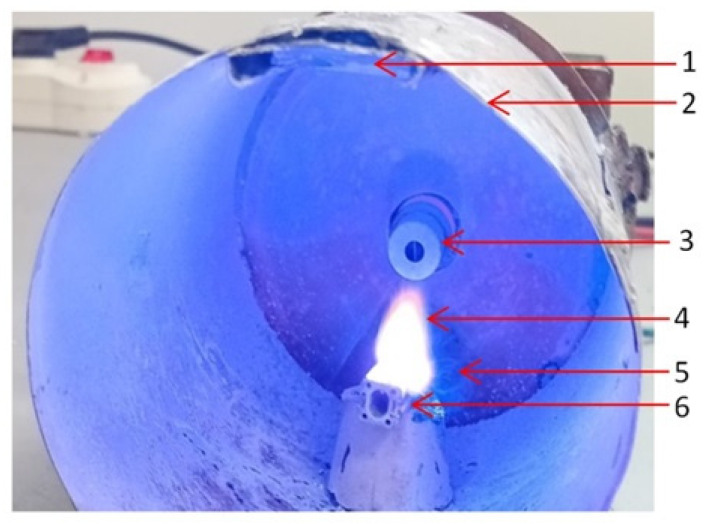
The Zn wire during vaporization process: 1—glass support, 2—waveguide, 3—magnetron antenna, 4—plasma, 5—metallic vapors, 6—ceramic support.

**Figure 12 ijms-26-08981-f012:**
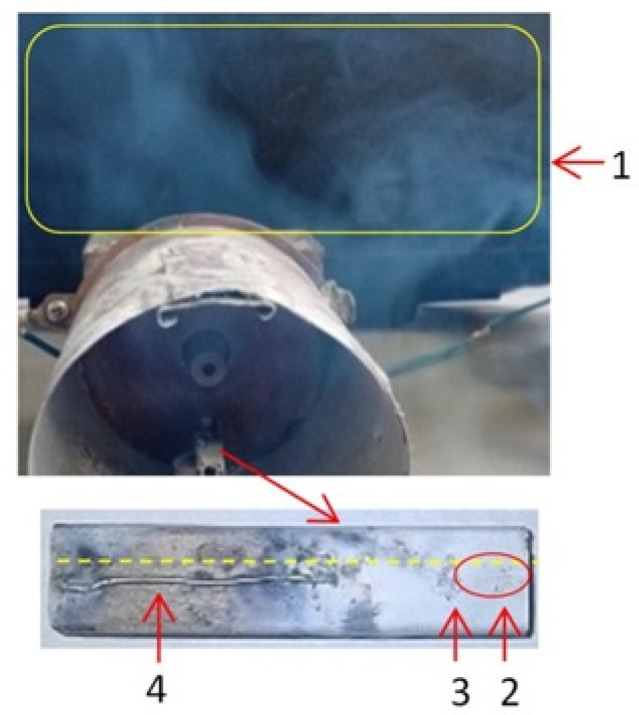
The Zn wire after vaporization process: 1—nanoparticles cloud, 2—high microwaves energy zone, 3—metallic oxide, 4—ceramic support, yellow line—the length of Zn wire before starting the microwaves.

**Table 1 ijms-26-08981-t001:** Emission Parameters of Microwave-Induced Zn Plasma.

ZnI	λ (m)	**g_Ki_ A_Ki_**	**E_k_**	ZnII	**λ (m)**	**g_Ki_ A_Ki_**	**E_k_**
	3.30 × 10^−7^	6.00 × 10^8^	7.782738		6.02 × 10^−7^	3.00 × 10^8^	14.62633
	6.36 × 10^−7^	2.40 × 10^8^	7.743871		6.10 × 10^−7^	3.73 × 10^8^	14.62934
	2.78 × 10^−7^	1.38 × 10^8^	8.502841		2.78 × 10^−7^	8.74 × 10^6^	12.56777
	4.81 × 10^−7^	3.50 × 10^8^	6.65451		2.76 × 10^−7^	7.64 × 10^6^	12.59821
	2.80 × 10^−7^	3.02 × 10^8^	8.503142		7.61 × 10^−7^	6.44 × 10^7^	14.19595
	3.28 × 10^−7^	2.70 × 10^8^	7.782333		7.76 × 10^−7^	1.30 × 10^8^	14.19595

**Table 2 ijms-26-08981-t002:** The dimension of the ZnO nanoparticles.

Sample		d_min_–d_max_ (nm)	FWHM (nm)	Mean ± SD(nm)
ZnO	NPs	12–63	15–32	24±9
	NRDs	58–354	96–218	155±57

## Data Availability

The original contributions presented in this study are included in the article. Further inquiries can be directed to the corresponding authors.

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
