# Peer review of "Direct High-Power Microwave Interaction with a Zinc Wire: A Novel Route to Crystalline ZnO Nanopowders Synthesis"

_ijms, 2025, doi:10.3390/ijms26188981_

Round 1
Reviewer 1 Report
Comments and Suggestions for Authors
The manuscript has been reviewed, which involved the preparation of nano zinc oxide using microwaves and elemental zinc. However, I find that the work requires additions, and the researchers should address the following questions.
-
This technology is not novel, as ZnO nanopowder was previously made using zinc powder (see ref. https://doi.org/10.3390/nano9070942).
Therefore, the word "novel" should be eliminated from the paper.
2. This preparation method has the potential to contaminate with deadly zinc vapor; how was this addressed?
3. The method utilized to prepare zinc oxide is not regarded economical, and there are cheaper and safer means of contamination, such as CBD.
4. What was the yield of nano zinc oxide compared to the amount of zinc used?
Author Response
Dear Reviewer 1,
Thank you for reviewing the article " Direct High Power Microwave Interaction with a Zinc Wire: A Novel Route to Crystalline ZnO Nanopowders Synthesis". Below I send you the answers to your questions.
The manuscript has been reviewed, which involved the preparation of nano zinc oxide using microwaves and elemental zinc. However, I find that the work requires additions, and the researchers should address the following questions.
Question 1:
This technology is not novel, as ZnO nanopowder was previously made using zinc powder (see ref. https://doi.org/10.3390/nano9070942).
Our answer: Lines 86-97 Therefore, this paper introduces a novel, low-cost, high-yield microwave-assisted synthesis method for producing ZnO nanoparticles. The method operates under ambient air conditions eliminating the need for chemical precursors, solvents and process gas and offers a non-contact, plasma-based route for nanoparticle generation.
The fundamental distinction between conventional microwave-assisted synthesis and the method described in this work lies in the mechanism by which plasma is generated.
In traditional systems, (including the paper : https://doi.org/10.3390/nano9070942 reference [50]) microwaves interact with a process gas to generate plasma, followed by the introduction of a solid-phase precursor into the plasma zone. By contrast, the new approach employs the direct interaction between microwaves and a metal wire,
allowing for the in situ synthesis of metal oxides directly from elemental metals.
Lines 327-347 In recent years, advanced systems have emerged for nanomaterial synthesis,
enabling the production of nanoparticles from chemical precursors or metallic powders
through the generation of hot plasma (Te ~6500K and ne = 1013 cm-3 [50]) in a process gas [48.49].
In these setups, the raw material is injected into the plasma, vaporized, and then condensed into nanoparticles. In contrast, the microwave synthesis method presented here distinguishes itself through a remarkably simple experimental setup, excellent energy efficiency, and a significantly shorter reaction time. It operates without the need for controlled environments or carrier gases.
Figure 12 shows a cloud of ZnO nanoparticles generated after just 10 seconds of microwave exposure at 650 W, during which approximately 42 mg of ZnO was generated. The structural characteristics of the ZnO nanoparticles, confirmed by TEM, SEM, and XRD analyses, indicate that their morphology is comparable to those produced by conventional microwave-based methods.
However, the key advantage of the present technique lies in its significantly higher production yield, approximately 15 grams/hour, compared to the 4 grams/hour typically achieved using conventional systems [50].
The significantly higher yield is primarily due to the elevated plasma temperature [51] achieved by this technique, reaching approximately 11,600 K and ne = 5.9 1016 cm-3.
This high temperature arises from the system’s unique operating principle, instead of relying on the interaction between microwaves and a process gas, as in conventional methods, the microwaves in this setup interact directly with a zinc wire in ambient air. Without the presence of the metal wire, plasma ignition does not occur under atmospheric conditions.
Therefore, the word "novel" should be eliminated from the paper.
Our answer : Our work is original because the synthesis is performed by directly exposing a zinc wire to microwave irradiation in ambient air, unlike conventional methods that require a process gas.
This unique route enables ZnO formation without any carrier or process gas, which, to our knowledge, has not been previously reported.
Question 2: This preparation method has the potential to contaminate with deadly zinc vapor; how was this addressed?
Our answer: In figure 9 (Materials and Methods) is displayed the design of the microwave device, the microwave deice is placed in a pressure chamber (1).
Lines 470-480: Upon microwave irradiation, the Zn wire rapidly vaporized as a result of localized plasma generation, producing Zn vapor that subsequently oxidized in ambient air and condensed both on the inner walls of the waveguide and on a vertically mounted glass substrate positioned above the plasma region.
The microwave plasma generator operates within a sealed pressure chamber, where the zinc wire is vaporized and subsequently oxidized to form ZnO nanoparticles. Only after nanoparticle condensation is complete is the chamber opened to collect the deposited substrate, ensuring safe operation without exposure to zinc vapor.
Question 3. The method utilized to prepare zinc oxide is not regarded economical, and there are cheaper and safer means of contamination, such as CBD.
Our answer: We respectfully disagree with the Reviewer 1 concern regarding the economic aspect.
The microwave generator employed in our method was assembled from commercially available components, without the need for specialized or costly equipment. Furthermore, the system is fully enclosed, ensuring safe operation. Unlike conventional plasma-based techniques, our approach does not require rare or high-purity process gases, which significantly reduces both cost and complexity. Therefore, the method can be considered both economical and safe.
Question 4. What was the yield of nano zinc oxide compared to the amount of zinc used?
Our answer: Lines 420-427 Regarding the Zn wire vaporization yield: the Zn wire from waveguide focal point (~1 cm) were vaporized cleanly with no residue. Once that segment had been consumed and the vaporization front moved away from the focal spot, localized melting occurred and small molten beads formed along the wire and at the edge of the ceramic support. Therefore, this nanoparticle synthesis method is limited by the absence of a mechanical system to feed the metal wire from a spool into the waveguide’s focal region and maintain the plasma discharge. Implementing such a system would allow for a continuous nanoparticle production.
Question 5. Please add uv-vis absorbance and estimate the band gap.
Our answer: The UV-VIS characterization and estimate the band gap was added.
Lines 204-228 2.4 Optical characterization of the ZnO nanoparticles
To evaluate the optical bandgap of ZnO nanoparticles produced by the direct interaction of microwaves with a Zn wire in atmospheric air, the nanoparticles were deposited onto a quartz substrate and their optical transmittance spectra (250nm - 850 nm) were recorded with a fiber-optic spectrometer (Ocean Optics USB2000) coupled to a DH-mini UV-Vis-NIR Deuterium-Halogen Light Source, then the bandgap was then determined by Tauc analysis. [41]. In Figure 3 is displayed the optical transmittance spectrum of the ZnO and in Figure 4 is displayed the Tauc plot for ZnO (direct-allowed transition)
Analyzing figure 4, it is observed that transmittance of the ZnO powder increases toward longer wavelengths; this trend is typical of strongly scattering media. The small fluctuations in the transmittance (Fig.4 ZnO raw) are consistent with a porous [42] and polydisperse powder layer.
The Tauc plot (direct-allowed) indicates an Eg~ 3.1eV band gap for the porous ZnO layer, which is lower than pure ZnO (Eg = 3.2eV-3.4eV). This lower value caused by sub-gap absorption from defects (oxygen vacancies, Zn interstitials) and strong scattering in the porous nanopowder, which broaden the absorption edge and bias the Tauc fit downward [43].
Question 6. From SEM image (Fig.3) the prepared sample contains more than structure such as nanoparticles, rods, and sheets), how the authors estimate particle size distribution (Fig.3)?
Our answer: Lines 239-241 Quantitative analysis of particle size distribution was conducted by evaluating approximately 200 individual NPs and NRDs from SEM images using ImageJ software (NIH). The resulting size histograms (Figure 3) were best fitted with a Gaussian function.
Lines 403-407 Statistical analysis of approximately 200 features from SEM images revealed that the ZnO sample consisted predominantly of nanorod-like (NRD) domains. The morphological yield of NRDs was ~87% (155 ± 57 out of 179 features), while nanoparticles accounted for only ~13% (24 ± 9 out of 179). These results demonstrate that the synthesis strongly promotes anisotropic growth along the c-axis, favoring the preferential formation of nanorods.
Question 7. The shape is not spherical; how did you apply k value of 0.93 in the Scherrer equation?
Our answer: The observation of Reviewer 1 regarding the shape is correct, since the crystallites are not spherical. We have modified the analysis of the crystallite size based on Scherrer equation, by considering a rod-like shape for crystallites (k = 1), rather than spherical.
Lines : 263 - 272 The crystal quality was evaluated considering the size of the crystalline domains (mean crystallite size), τ by the well-known Scherrer equation (5), that gives the relation between the crystalline domain size and peak broadening, b: , where: k is a dimensionless factor that represents the shape factor of the crystallites and it varies between 0.62 and 2.08 depending on the crystallite geometry and distribution [44] and θ is the angular position of the diffraction peak. Based on the analysis of the (101) XRD diffraction peak and the application of the Scherrer equation (Equation 4), in the assumption of rod-like shape for crystallite shape [45] the average crystallite size of ZnO was determined to be approximately 25.6 nm, which is very similar to the diameter of the crystals estimated from SEM images.
Question 8. Wide range of FTIR (500-4000 cm-1) is required.
Our answer: Lines 302-305 Figure 8(a), recorded in the 4000 cm-1 - 380 cm-1 range provides the complete FTIR spectrum of ZnO deposited on the glass substrate, while Figure 8(b) presents an enlarged view of the 500 cm-1 - 380 cm-1 region, highlighting the characteristic Zn- O peaks.
Question 9. Please add the EDX map to explain the Zn-O distribution in the sample.
Our answer: EDX elemental maps of the ZnO nanostructured layer was added (Figure 5)
Lines 237 – 239 Elemental composition was evaluated using energy-dispersive X-ray spectroscopy (EDX, Figure 5) mapping, which identified zinc (Zn) and oxygen (O) as the primary constituents being uniformly distributed on the glass substrate.
Thank you for your consideration of this manuscript.
Sincerely,
Dr. Marian Mogildea

Reviewer 2 Report
Comments and Suggestions for Authors
This article reported the synthesis of zinc oxide (ZnO) nanopowders based on the direct interaction of high-power microwave radiation with a zinc wire in atmospheric air. However, the microwave-assisted growth of ZnO nanostructures has been documented ten years ago (Journal of Alloys and Compounds 2011, 509, 6859–6863; Langmuir 2010, 26, 8, 5976–5984). The novelty of this paper is missing. So, the present paper is not recommended to be accepted.
1 The summary of the previous works on this issue is necessary to highlight the novelty of this paper. The references should be updated.
2 In Figure 2 (b), Boltzmann Plot for Zn II, the data points are distributed at both ends. An explanation is necessary.
3 SEM analysis confirmed the presence of both nanoparticles (NPs) and nanorod-like domains (NRDs). What is the yield of ZnO nanorods?
4 TEM Images of the nanorod-like domains are missing.
5 The figures are not clear enough. High resolution images are required.
Author Response
Dear Reviewer 2,
Thank you for reviewing the article " Direct High Power Microwave Interaction with a Zinc Wire: A Novel Route to Crystalline ZnO Nanopowders Synthesis". Below I send you the answers to your questions.
This article reported the synthesis of zinc oxide (ZnO) nanopowders based on the direct interaction of high-power microwave radiation with a zinc wire in atmospheric air.
Question 1: However, the microwave-assisted growth of ZnO nanostructures has been documented ten years ago (Journal of Alloys and Compounds 2011, 509, 6859–6863; Langmuir 2010, 26, 8, 5976–5984). The novelty of this paper is missing. So, the present paper is not recommended to be accepted.
Our answer: Lines 86-97. Therefore, this paper introduces a novel, low-cost, high-yield microwave-assisted synthesis method for producing ZnO nanoparticles. The method operates under ambient air conditions eliminating the need for chemical precursors, solvents and process gas and offers a non-contact, plasma-based route for nanoparticle generation.
The fundamental distinction between conventional microwave-assisted synthesis and the method described in this work lies in the mechanism by which plasma is generated.
In traditional systems, (including the papers Journal of Alloys and Compounds 2011, 509, 6859–6863; Langmuir 2010, 26, 8, 5976–5984 references [48,49] microwaves interact with a process gas to generate plasma, followed by the introduction of a solid-phase precursor into the plasma zone.
By contrast, the new approach employs the direct interaction between microwaves and a metal wire, allowing for the in situ synthesis of metal oxides directly from elemental metals.
Lines 327-347 In recent years, advanced systems have emerged for nanomaterial synthesis, enabling the production of nanoparticles from chemical precursors or metallic powders through the generation of hot plasma (Te ~6500K and ne = 1013 cm-3 [50]) in a process gas.
In these setups, the raw material is injected into the plasma, vaporized, and then condensed into nanoparticles.
In contrast, the microwave synthesis method presented here distinguishes itself through a remarkably simple experimental setup, excellent energy efficiency, and a significantly shorter reaction time. It operates without the need for controlled environments or carrier gases.
Figure 10 shows a cloud of ZnO nanoparticles generated after just 10 seconds of microwave exposure at 650 W, during which approximately 42 mg of ZnO was generated. The structural characteristics of the ZnO nanoparticles, confirmed by TEM, SEM, and XRD analyses, indicate that their morphology is comparable to those produced by conventional microwave-based methods.
However, the key advantage of the present technique lies in its significantly higher production yield, approximately 15 grams/hour, compared to the 4 grams/hour typically achieved using conventional systems [50].
The significantly higher yield is primarily due to the elevated plasma temperature [51] achieved by this technique, reaching approximately 11,600 K and ne = 5.9 1016 cm-3.
This high temperature arises from the system’s unique operating principle, instead of relying on the interaction between microwaves and a process gas, as in conventional methods, the microwaves in this setup interact directly with a zinc wire in ambient air.
Without the presence of the metal wire, plasma ignition does not occur under atmospheric conditions.
Question 2: The summary of the previous works on this issue is necessary to highlight the novelty of this paper. The references should be updated.
Our answer: The references were updated. Lines 97-98 : This approach has enabled the successful synthesis of high-purity metal oxide nanoparticles, such as In2O3, WO3, and TiO2, directly from elemental metal wires [20-22].
Question 3: In Figure 2 (b), Boltzmann Plot for Zn II, the data points are distributed at both ends. An explanation is necessary.
Our answer: Lines 183-187 The clustering of data points at both ends of the Boltzmann plot for Zn II reflects the variable accuracy in measuring strong versus weak emission lines, primarily influenced by differences in the signal-to-noise ratio [38]. The surrounding nanoparticle cloud can attenuate the plasma emission through scattering and absorption [39], thereby reducing line intensity; nevertheless, the high electron temperature and the strong correlation coefficients (R2 > 0.9 for Zn I and Zn II) confirm the reliability of the results.
Question 4: SEM analysis confirmed the presence of both nanoparticles (NPs) and nanorod-like domains (NRDs). What is the yield of ZnO nanorods?
Our answer: Lines 403-407. Statistical analysis of approximately 200 features from SEM images revealed that the ZnO sample consisted predominantly of nanorod-like (NRD) domains. The morphological yield of NRDs was ~87% (155 ± 57 out of 179 features), while nanoparticles accounted for only ~13% (24 ± 9 out of 179). These results demonstrate that the synthesis strongly promotes anisotropic growth along the c-axis, favoring the preferential formation of nanorods.
Question 5: TEM Images of the nanorod-like domains are missing.
Our answer: We would like to clarify that TEM images of the nanorod-like domains are already included in the manuscript. Specifically, Figure 7(a) shows BFTEM images of the ZnO nanopowder at two different magnifications (100 nm scale), where the nanorod-like structures are clearly visible. To avoid any ambiguity, we have revised the figure legend to explicitly state that these TEM images highlight the nanorod domains, consistent with the SEM observations.
Question 6: The figures are not clear enough. High resolution images are required.
Our answer: Some figures have been improved, while others remained unchanged because they are generated directly by the measurement instrument. Any post-processing intervention could alter the integrity of the experimental results.
Thank you for your consideration of this manuscript.
Sincerely,
Dr. Marian Mogildea

Reviewer 3 Report
Comments and Suggestions for Authors
Authors should revise the manuscript and review the presented thesis. Idea to obtaine oxide materials via to bomb metal source by laser/plasma/ion-beam/microwave/super and ultra-high waves or so is not new and not well-controlled(sometimes dangerous). The paper needs to be strengthened before publishing at IJMS or any material-interested journal.
Abstract, conclusions and presented main idea- are different. All should be synchronized.
Please, provide or add inforamtion about quantuty of attemped. Is it only one sample were obtained by provided technique? Didi you repeat the synthesis? How about of repeatability of the method?
Please, provide the comparison results with other techniques or materials produced by similar method to understant where this work stay and how significatnt the result. It would be great for readers if it does.
Geometry of state (figure 7) including all distances and sized of details is important decribtion which should be add to experimental part and provide while the estimation and calculation of energy were showed.
Also the small missaunderstading should be meet by authors:
-"The resulting nanorods exhibit a uniform morphology..." should be revised. According to sem images morphoology far from uniform. It is better to describe the obtained materials and its morphology correctly.
-The X axis of Figure 6 shoud be noted. The resolution of plot should be presentable and understandable. Now the graph and discussion are not relevant to each other. Plaese use Infrared and Raman Characteristic Group Frequencies by G. Socrates instead of imagination. Laso should be described C-O bonds. It may be sensitive to explantion of mechanism of process.
-The XRD data base (for example ICSD) and number of card ZnO should be provided to correct identification of phase content. Please, provide xrd of glass without powders to indetify the widht peak around 20 grad.
-Please, give info which shape authors take when use shererre eq.? Which from obtained particles has correct and expected form?
Author Response
Dear Reviewer 3,
Thank you for reviewing the article " Direct High Power Microwave Interaction with a Zinc Wire: A Novel Route to Crystalline ZnO Nanopowders Synthesis". Below I send you the answers to your questions.
Authors should revise the manuscript and review the presented thesis.
Question 1: Idea to obtain oxide materials via to bomb metal source by laser/plasma/ion-beam/microwave/super and ultra-high waves or so is not new and not well-controlled (sometimes dangerous).
The paper needs to be strengthened before publishing at IJMS or any material-interested journal.
Our answer: In figure 9 (Materials and Methods) is displayed the design of the microwave device, the microwave deice is placed in a pressure chamber (1).
Lines 470-479. Upon microwave irradiation, the Zn wire rapidly vaporized as a result of localized plasma generation, producing Zn vapor that subsequently oxidized in ambient air and condensed both on the inner walls of the waveguide and on a vertically mounted glass substrate positioned above the plasma region. The microwave plasma generator operates within a sealed pressure chamber, where the zinc wire is vaporized and subsequently oxidized to form ZnO nanoparticles. Only after nanoparticle condensation is complete is the chamber opened to collect the deposited substrate, ensuring safe operation without exposure to zinc vapor.
Question 2: Abstract, conclusions and presented main idea - are different. All should be synchronized.
Our answer: The conclusions were synchronized with abstract and main idea.
Question 3: Please, provide or add information about quantity of attempted. Is it only one sample was obtained by provided technique? Didi you repeat the synthesis? How about of repeatability of the method?
Our answer: Lines 414-418. To assess reproducibility, the synthesis was repeated in more than 40 independent deposition cycles carried out for different experimental purposes. In every case the process produced ZnO nanorods with consistent morphology, size distribution, and crystallinity. For this study, two representative samples were selected for detailed characterization. The uniform behavior of the synthesized ZnO across all cycles demonstrates that the method is highly reliable and readily reproducible.
Question 4: Please, provide the comparison results with other techniques or materials produced by similar method to understand where these works stay and how significant the result. It would be great for readers if it does.
Our answer: Lines 86-97 Therefore, this paper introduces a novel, low-cost, high-yield microwave-assisted synthesis method for producing ZnO nanoparticles. The method operates under ambient air conditions eliminating the need for chemical precursors, solvents and process gas and offers a non-contact, plasma-based route for nanoparticle generation.
The fundamental distinction between conventional microwave-assisted synthesis and the method described in this work lies in the mechanism by which plasma is generated.
In traditional systems, microwaves interact with a process gas to generate plasma, followed by the introduction of a solid-phase precursor into the plasma zone [48,49,50].
By contrast, the new approach employs the direct interaction between microwaves and a metal wire, allowing for the in situ synthesis of metal oxides directly from elemental metals.
Lines 327-347 In recent years, advanced systems have emerged for nanomaterial synthesis, enabling the production of nanoparticles from chemical precursors or metallic powders through the generation of hot plasma (Te ~6500K and ne = 1013 cm-3 [50]) in a process gas.
In these setups, the raw material is injected into the plasma, vaporized, and then condensed into nanoparticles.
In contrast, the microwave synthesis method presented here distinguishes itself through a remarkably simple experimental setup, excellent energy efficiency, and a significantly shorter reaction time. It operates without the need for controlled environments or carrier gases.
Figure 10 shows a cloud of ZnO nanoparticles generated after just 10 seconds of microwave exposure at 650 W, during which approximately 42 mg of ZnO was generated. The structural characteristics of the ZnO nanoparticles, confirmed by TEM, SEM, and XRD analyses, indicate that their morphology is comparable to those produced by conventional microwave-based methods.
However, the key advantage of the present technique lies in its significantly higher production yield, approximately 15 grams/hour, compared to the 4 grams/hour typically achieved using conventional systems [47].
The significantly higher yield is primarily due to the elevated plasma temperature [51] achieved by this technique, reaching approximately 11,600 K and ne = 5.9 1016 cm-3.
This high temperature arises from the system’s unique operating principle, instead of relying on the interaction between microwaves and a process gas, as in conventional methods, the microwaves in this setup interact directly with a zinc wire in ambient air. Without the presence of the metal wire, plasma ignition does not occur under atmospheric conditions.
Question 5: Geometry of state (figure 9) including all distances and sized of details is important decribtion which should be add to experimental part and provide while the estimation and calculation of energy were showed.
Our answer: Lines 413-434 The system utilizes a magnetron (frequency = 2.45GHz, power of the microwaves = 800W) coupled to a TM011 cylindrical copper waveguide (10.5 cm diameter × 11 cm length), optimized to generate a high-intensity microwave field (~8 MW/cm2 at its first resonant node, located on the cavity’s cylindrical axe at 6.5 cm away from the antenna. Figure 9 shows the schematic of the setup.
Also the small missaunderstading should be meet by authors:
Question 5: "The resulting nanorods exhibit a uniform morphology..." should be revised. According to sem images morphoology far from uniform. It is better to describe the obtained materials and its morphology correctly.
Our answer: the text was corrected.
Question 6: The X axis of Figure 6 shoud be noted.
Our answer: The X axis was added.
Question 7: The resolution of plot should be presentable and understandable.
Our answer: The figure was improved.
Question 8: Now the graph and discussion are not relevant to each other.
Please use Infrared and Raman Characteristic Group Frequencies by G. Socrates instead of imagination.
Our answer: Lines 314-316 The FTIR spectrum of the ZnO sample deposited on glass exhibits absorption bands that correspond well with the characteristic group frequencies described in Infrared and Raman Characteristic Group Frequencies by G. Socrates [46], confirming the reliability of the spectral assignments.
Question 9: Also should be described C-O bonds. It may be sensitive to explantion of mechanism of process.
Our answer: Lines 318-321. No organic contamination bands or absorption bands corresponding to C-O stretching vibrations were detected in the FTIR spectrum. This absence is expected, since the synthesis involved only the direct oxidation of metallic zinc wire under microwave irradiation in ambient air, without any carbon-containing precursors. The result confirms that the obtained ZnO nanostructures are free of carbon-oxygen impurities.
Question 10: The XRD data base (for example ICSD) and number of card ZnO should be provided to correct identification of phase content.
Our answer: The figure 6 was improved, the ICDD was added.
Question 11: Please, provide xrd of glass without powders to indetify the widht peak around 20 grad.
Our answer: The XRD of glass was added (Figure 6)
Question 12: Please, give info which shape authors take when use shererre eq.?
Which from obtained particles has correct and expected form?
Lines : 236-272 The crystal quality was evaluated considering the size of the crystalline domains (mean crystallite size), τ by the well-known Scherrer equation (5), that gives the relation between the crystalline domain size and peak broadening, b: , where: k is a dimensionless factor that represents the shape factor of the crystallites and it varies between 0.62 and 2.08 depending on the crystallite geometry and distribution [41] and θ is the angular position of the diffraction peak. Based on the analysis of the (101) XRD diffraction peak and the application of the Scherrer equation (Equation 4), in the assumption of rod-like shape for crystallite shape [42] the average crystallite size of ZnO was determined to be approximately 25.6 nm, which is very similar to the diameter of the crystals estimated from SEM images.
Thank you for your consideration of this manuscript.
Sincerely,
Dr. Marian Mogildea

Round 2
Reviewer 1 Report
Comments and Suggestions for Authors
The revised copy of the manuscript is suitable for publishing.
Reviewer 2 Report
Comments and Suggestions for Authors
The responses to reviewers’ comments seems to be reasonable. I am happy with the corrections and would like to recommend it for publication.